# The Effect of Antioxidant Polyphenol Supplementation on Cardiometabolic Risk Factors: A Systematic Review and Meta-Analysis

**DOI:** 10.3390/nu16234206

**Published:** 2024-12-05

**Authors:** Sitong Wan, Junjie Luo, Yinhua Zhu, Peng An, Yongting Luo, Qingchang Xing

**Affiliations:** 1Department of Nutrition and Health, China Agricultural University, Beijing 100193, China; adeline7wan@163.com (S.W.); luojj@cau.edu.cn (J.L.); zhuyinhua@cau.edu.cn (Y.Z.); anpeng@cau.edu.cn (P.A.); 2Chinese People’s Liberation Army (PLA) Medical School, Beijing 100091, China

**Keywords:** antioxidant polyphenol, cardiovascular disease, meta-analysis

## Abstract

**Background**: Polyphenol is considered to exert a favorable impact on cardiovascular health. **Methods**: To summarize the role of polyphenol antioxidant supplements in cardiovascular disease, we searched for randomized controlled trials up to 10th November 2024 that reported estimates of the effects of polyphenol antioxidant supplements on cardiometabolic risk factors. **Results**: Of the 17,126 participants in the 281 studies, weighted mean differences [95% confidence intervals] were derived for the intervention condition utilizing random effects modeling. Our results suggest that multiple polyphenol supplements improved cardiovascular risk markers in the overall population. For example, catechin supplementation decreased systolic (−1.56 [−2.75, −0.37] mmHg) and diastolic blood pressure (−0.95 [−1.69, −0.20] mmHg), anthocyanin supplementation improved multiple blood lipid profiles, and curcumin supplementation benefited indicators of glucose metabolism. **Conclusions**: Our meta-analysis provides comprehensive evidence that antioxidant polyphenol supplementation can have beneficial effects on various cardiometabolic risk factors in the general population. The observed improvements in blood pressure, lipid profile, and glycemic status support the potential role of these supplements in cardiovascular health promotion. However, the heterogeneity among studies indicates that more research is needed to fully understand the optimal use of different polyphenols. Future research should concentrate on conducting a greater number of well-designed randomized controlled trials over extended periods of time to evaluate the long-term impact on cardiovascular endpoints and to ascertain the optimal doses and durations of antioxidant polyphenol supplementation. Furthermore, additional research is required to gain a deeper understanding of the underlying mechanisms responsible for these cardioprotective effects.

## 1. Introduction

A substantial proportion of mortality and morbidity related to cardiovascular diseases (CVD) and type 2 diabetes (T2D) can be attributed to suboptimal dietary patterns [1,2,3]. Adequate consumption of fruits and vegetables constitutes a vital part of a healthy diet regime. It has been calculated that diets deficient in fruits and vegetables account for 1.5–4% of the global disease burden [4,5,6]. The Dietary Guidelines for Americans (2020–2025) recommend for citizens to increase their fruit and vegetable intake as a means of reducing the risk of CVD [7,8]. In addition to vegetables and fruits, beverages such as tea, cocoa, and coffee have also been linked to a reduced risk of CVD [9,10,11]. A common feature of these foods is that they all include an abundance of antioxidant polyphenols.

Polyphenols, which are phytochemicals or secondary plant compounds, are non-essential plant nutrients existing in plants and algae. In these organisms, they serve as a natural defense mechanism against UV radiation, inflammation, and consumption by herbivores [12]. Polyphenols are usually classified as flavonoids and non-flavonoids, with the flavonoids commonly found in foods being anthocyanins, flavanols, flavonoids, phenolic acids, isoflavones, flavanones, and astragals [12,13]. Flavonoids exhibit a range of biological activities, including antioxidant, anti-inflammatory, anticancer, and cardiovascular protective effects. Their antioxidant properties are closely related to their structural features, with the presence of multiple hydroxyl groups enhancing their antioxidant potential. Additionally, flavonoids have been shown to have a variety of pharmacological effects, including antidepressant, anti-tumor, and anti-osteoporosis properties [12,13]. Non-flavonoids encompass a range of compounds, including phenolic acids, particularly hydroxybenzoic acid and cinnamic acid, which have been demonstrated to possess biological activity [14,15].

Epidemiological studies have demonstrated that the risk of CVD is reduced by 46 percent in individuals who consume a diet rich in polyphenols [16,17]. Furthermore, studies have demonstrated that polyphenols facilitate the enhancement of endothelial function, the prevention of aberrant platelet aggregation, the reduction of inflammation, and the improvement of plasma lipid profiles, all of which contribute to the promotion of cardiovascular health. However, the precise mechanisms through which polyphenols exert these effects remain to be fully elucidated [18,19].

Despite the existing evidence suggesting the potential benefits of polyphenols on cardiovascular health, there are still significant inconsistencies in the literature. This is of great clinical importance to gain a deeper understanding of the impact of antioxidant polyphenol supplementation on cardiometabolic risk factors. If proven effective, polyphenol supplements could represent a relatively inexpensive and accessible preventive or adjunctive treatment option for cardiovascular diseases. The objective of our review was to synthesize the available evidence in a more comprehensive and systematic manner, with the aim of addressing the aforementioned uncertainties and providing a more definitive understanding of the role of antioxidant polyphenol supplementation in cardiovascular health.

To pool the available evidence regarding the role of antioxidant polyphenol supplementation in advancing cardiovascular health, a systematic review and meta-analysis was conducted that included all available randomized controlled trials of antioxidant polyphenol supplementation and its effect on CVD risk factors. This comprehensive approach provides a more holistic view of the potential benefits of antioxidant polyphenols as a class of compounds. The impetus behind conducting this review was to provide evidence-based recommendations that could potentially be translated into clinical practice and public health strategies, with the aim of improving cardiovascular health outcomes.

## 2. Methods

Adhering to the Preferred Reporting Items for Systematic Reviews and Meta-Analyses guidelines [20], the study protocol was entered into the International Registry of Systematic Reviews (PROSPERO: CRD420245555557405). Ethical approval for each of the included randomized trials was granted by the corresponding institutional review board.

### 2.1. Search Strategy

RCTs published up to 10th November 2024 were searched in the PubMed, Web of Science, and Embase databases. The following terms were used in the systematic search: (“polyphenol” OR “flavonoid” OR “flavone” OR “flavanone” OR “flavanol” OR “Isoflavone” OR “isoflavonoid” OR “anthocyanin” OR “proanthocyanidin” OR “lignans” OR “enterolignan” OR “curcumin” OR “catechin” OR “silymarin” OR “kaempferol” OR “quercetin” OR “resveratrol” OR “genistein” OR “secoisolariciresinol” OR “myricetin” OR “lariciresinol” OR “pinoresinol” OR “tamarixetin” OR “apigenin” OR “polyphenolacid” OR “pycnogenol” OR “epicatechin” OR “Epigallocatechin gallate” OR “EGCG” OR “Hesperidin” OR “gallic acid” OR “chlorogenic acid” OR “cinnamic acid” OR “caffeic acid” OR “ferulic acid” OR “coumaric acid” OR “erucic acid”) AND (“Blood glucose” OR “Blood sugar” OR “glycemic” OR “Blood lipids” OR “triglyceride” OR “cholesterol” OR “Blood pressure”).

### 2.2. Eligibility and Ineligibility Criteria

The extended analysis encompassed an assessment of how supplementing with antioxidant functional lipids impacted various health parameters. These included cardiovascular measures (systolic blood pressure [SBP] and diastolic blood pressure [DBP]), blood lipid profiles (total cholesterol [TC], high-density lipoprotein cholesterol [HDL-C], low-density lipoprotein cholesterol [LDL-C], and triglycerides [TG]), and indicators of glucose metabolism (fasting blood glucose [FBG], hemoglobin [A1C], and fasting blood insulin [FBI]).

We excluded trials lacking randomization, pertinent cardiovascular metabolic outcomes, placebo or matched controls, or mean alterations and standard deviations (SD). Furthermore, studies whose intervention duration was less than one week or in which participants at baseline had a major cardiovascular disease, psychiatric disorder, or any other serious medical condition were excluded.

### 2.3. Selection of Study

Two reviewers conducted a comprehensive and meticulous examination of the existing literature to ensure the accuracy and completeness of the review. Subsequently, the reviewers undertook a comprehensive examination of each article. The titles and abstracts were then examined in accordance with a set of pre-established criteria to determine which ones warranted a more comprehensive analysis. A third reviewer was then engaged to examine the selected studies in detail. This reviewer sought to identify any errors or inconsistencies that might have been overlooked in the initial review. If any issues were identified, the third reviewer facilitated panel discussions. All three reviewers engaged in detailed discussions and shared their opinions. The discussions were guided by the predefined criteria to reach a consensus on including or excluding each study.

### 2.4. Extraction of Data

The extraction of data from selected studies encompassed the lead author’s name, publication date, study population location, research design, characteristic of participants (participants’ number, average age, sex, and physical condition), substance of the intervention, control, dose of the intervention, and duration of the intervention. SD_change_ = square root [(SD_baseline_^2^ + SD_endpoint_^2^)/2] was calculated to estimate the SD of the mean variance if there were no reported SD values for the mean difference between baseline cardiac risk factors and the endpoint [21]. When the outcome data was presented as a graph, WebPlotDigitizer (https://automeris.io/WebPlotDigitizer/, version 4.8, accessed on 1 January 2024) was used to estimate the values.

### 2.5. Assessment of Quality

Included trials were assessed for risk of bias according to the recommendations of the Cochrane Collaboration Handbook [22], including six aspects: bias in selection involves creating a random sequence and hiding the allocation; bias toward performance involves blinding participants and staff; disclosure bias involves blinding outcome evaluations; bias due to attrition involves incomplete outcome data; bias in reporting involves voluntary reporting; and other biases are present.

To assess the quality of evidence, researchers employed the GRADE (Grading of Recommendations, Assessment, Development, and Evaluation) approach. This method classifies evidence quality into four levels: high, moderate, low, or very low [23]. Initial GRADE quality scores defaulted to high before being downgraded according to prespecified domains, including risk of bias (more than 20%), inconsistency (*I*^2^ > 50% and *P_heterogeneity_* < 0.1), indirectness (limitations to the generalization of the results), imprecision (overlap of the 95% confidence interval (CI) with the smallest significant difference, i.e., blood pressure of 2 mmHg [24], blood lipids of 0.1 mol/L [25], A1C of 0.5% [26], FBG of 0.56 mmol/L or 10 mg/dL [27], and FBI of 5 pmol/L [28]), and bias in publications (strong proof in the effects of small studies).

### 2.6. Statistical Analysis

Each study was compared using the mean differences and standard deviations for specific cardiometabolic risk variables between the intervention and control groups. Effect sizes were assessed according to the Cochrane guidelines [21]. Effects sizes for cardiometabolic risk factors were generated from random effects models presented as weighted mean differences and 95% CIs. *I*^2^ statistics were used to estimate heterogeneity among the included studies. Significance of variance was assigned at *p* < 0.05, with *I*^2^ > 50% indicating evidence of significant variance [29]. Subgroup analyses were conducted in accordance with the varying levels of cardiometabolic health observed among the trial participants. Egger’s regression-based assay and eye inspection of funnel plots were used to evaluate the likelihood of publication bias (*p* < 0.05 shows publication bias [30]). To evaluate the impact of specific research upon the aggregate effect size, sensitivity analyses were performed by excluding one study at a time [31]. The software RevMan (version 5.4) and the software Stata/SE (version 17.0) were used in all of the statistical analyses.

## 3. Results

### 3.1. The Process of Selecting Studies for Inclusion and the Characteristics of the Studies

Figure 1 shows a flow diagram of the recruitment process in this study. Altogether, 3683 articles were retrieved through an exhaustive search. Among the remaining 1132 unduplicated articles, 543 articles were defined as irrelevant based on titles and abstracts, while 589 articles were assessed for relevance. Studies without randomization (*n* = 102) or pertinent outcomes (*n* = 129), with a short intervention duration (*n* = 45), or including participants with severe disorders (*n* = 89) were excluded, resulting in 224 articles (284 studies) left for data extraction for further analysis. The characteristics of the included trials are outlined in the Appendix A for readers’ convenience.

A total of 284 studies with, altogether, 17,613 participants aged between 18 and 75 years (median age: 52.2 years) were included in the current analysis. The included studies were performed on participants who took anthocyanins (*n* = 43), catechin (*n* = 42), chlorogenic acid (*n* = 27), curcumin (*n* = 24), flavanol (*n* = 27), flavonoid (*n* = 13), gallic acid (*n* = 7), genistein (*n* = 11), hesperidin (*n* = 7), isoflavone (*n* = 32), quercetin (*n* = 19), and resveratrol (*n* = 30).

### 3.2. The Impact of Antioxidant Polyphenol Supplementation on Blood Pressure

To validate how antioxidant polyphenol supplementation affects blood pressure 191 relevant RCTs involving 10,703 participants were incorporated (Figure 2). In the total population, supplementation with catechin (SBP: −1.56 [−2.75, −0.37] mmHg; DBP: −0.95 [−1.69, −0.20] mmHg), genistein (SBP: −10.02 [−11.55, −8.49] mmHg; DBP: −9.13 [−12.80, −5.46] mmHg), and resveratrol (SBP: −3.25 [−6.03, −0.48] mmHg; DBP: −2.32 [−4.07, −0.57] mmHg) significantly improved both SBP and DBP. Curcumin (SBP: −1.42 [−2.56, −0.28] mmHg), flavanol (SBP: −1.47 [−2.89, −0.06] mmHg), and quercetin (SBP: −1.38 [−2.63, −0.13] mmHg) supplementation significantly decreased SBP in the total population. Additionally, flavonoid supplementation significantly improved DBP (−1.68 [−3.34, −0.03] mmHg).

### 3.3. The Impact of Antioxidant Polyphenol Supplementation on Blood Lipids

To evaluate how antioxidant polyphenol supplementation affects blood lipids, 221 relevant RCTs comprising 14,076 participants were included (Figure 3). Across the entire population, anthocyanin supplementation significantly improved all blood lipid profiles, including LDL-C (−0.18 [−0.31, −0.06] mmol/L), HDL-C (0.18 [0.12, 0.25] mmol/L), TC (−0.18 [−0.33, −0.02] mmol/L), and TG (−0.47 [−0.70, −0.24] mmol/L). Chlorogenic acid supplementation significantly decreased LDL-C (−0.24 [−0.38, −0.11] mmol/L), TC (−0.39 [−0.62, −0.16] mmol/L), and TG (−0.10 [−0.15, −0.04] mmol/L). Flavonoid supplementation significantly improved LDL-C (−0.32 [−0.61, −0.04] mmol/L), HDL-C (0.15 [0.02, 0.29] mmol/L), and TG (−0.70 [−1.37, −0.03] mmol/L). Genistein supplementation significantly decreased LDL-C (−0.43 [−0.81, −0.04] mmol/L) and TC (−0.22 [−0.35, −0.08] mmol/L). Additionally, curcumin (0.39 [0.22, 0.56] mmol/L) and flavanol (0.09 [0.04, 0.13] mmol/L) supplementation significantly improved HDL-C in the total population.

### 3.4. The Impact of Antioxidant Polyphenol Supplementation on Glycemic Status

There were 164 relevant RCTs, comprising 10,961 participants, to estimate how antioxidant polyphenol supplementation affects glycemic status (Figure 4). In the total population, curcumin supplementation significantly improved all glycemic parameters, including FBG (−0.43 [−0.68, −0.19] mmolL), FBI (−10.14 [−14.13, −6.14] pmol/L), and A1C (−0.49 [−0.83, −0.14]%). Chlorogenic acid (FBG: −0.16 [−0.27, −0.06] mmol/L; FBI: −5.36 [−9.62, −1.10] pmol/L), flavanol (FBG: −0.17 [−0.30, −0.03] mmol/L; FBI: −14.86 [−21.02, −8.71] pmol/L), and genistein (FBG: −0.44 [−0.52, −0.37] mmol/L; FBI: −11.61 [−15.40, −7.82] pmol/L) supplementation significantly improved FBG and FBI. Catechin supplementation significantly decreased FBG (−0.10 [−0.18, −0.03] mmol/L) and A1C (−0.12 [−0.23, −0.02]%). As well, quercetin (−8.09 [−15.53, −0.66] pmol/L) and resveratrol (−2.84 [−5.61, −0.06] pmol/L) supplementation significantly decreased FBI, while anthocyanin (−0.09 [−0.17, −0.02] mmol/L) supplementation significantly improved FBG in the total population.

### 3.5. The Impact of Antioxidant Polyphenol Supplementation on Participants with Varying Cardiometabolic Health Statuses

Antioxidant polyphenol supplementation subgroup analysis was carried out among participants with diverse cardiometabolic health conditions (Figure 5). Among a healthy population, antioxidant polyphenol supplementation significantly bettered DBP (−0.68 [−1.33, −0.03] mmHg), HDL-C (0.07 [0.02, 0.11] mmol/L), TG (−0.13 [−0.22, −0.03] mmol/L), FBG (−0.14 [−0.24, −0.04] mmol/L), and FBI (−6.01 [−9.19, −2.83] pmol/L). For participants with pre-diabetes or T2D, antioxidant polyphenol supplementation significantly improved SBP (−4.54 [−7.14, −1.93] mmHg), HDL-C (0.13 [0.06, 0.19] mmol/L), TC (−0.32 [−0.49, −0.15] mmol/L), TG (−0.32 [−0.53, −0.12] mmol/L), FBG (−0.43 [−0.62, −0.24] mmol/L), and A1C (−0.19 [−0.35, −0.03]%). For hypertensive participants, antioxidant polyphenol supplementation improved blood pressure, including SBP (−2.37 [−3.65, −1.09] mmHg) and DBP (−1.13 [−1.92, −0.32] mmHg). For dyslipidemia participants, antioxidant polyphenol supplementations improved LDL-C (−0.52 [−0.85, −0.19] mmol/L), HDL-C (0.12 [0.03, 0.20] mmol/L), and TC (−0.47 [−0.83, −0.10] mmol/L). For participants who were overweight or obese, antioxidant polyphenol supplementation improved TC (−0.18 [−0.33, −0.02] mmol/L), TG (−0.09 [−0.18, −0.00] mmol/L), FBG (−0.06 [−0.11, −0.02] mmol/L), and FBI (−1.67 [−3.19, −0.16] pmol/L). For participants with metabolic syndrome, antioxidant polyphenol supplementation improved all lipid profiles, including LDL-C (−0.23 [−0.40, −0.05] mmol/L), HDL-C (0.03 [0.00, 0.06] mmol/L), TC (−0.17 [−0.32, −0.02] mmol/L), and TG (−0.17 [−0.30, −0.03] mmol/L). Lastly, antioxidant polyphenol supplementation improved SBP (−0.85 [−1.68, −0.02] mmHg) in postmenopausal women.

## 4. Discussion

The point of this study was to comprehensively analyze and select suitable antioxidant polyphenol supplements for optimizing cardiometabolic health (Figure 6). The findings indicate that a number of polyphenols, including genistein, anthocyanin, catechin, chlorogenic acid, curcumin, flavanol, flavonoid, resveratrol, and quercetin, have the potential to significantly enhance the management of cardiovascular disease risk factors, such as elevated blood pressure, dyslipidemia, and impaired glucose metabolism, across the general population. The findings indicated that antioxidant polyphenol supplementation was beneficial for participants with varying levels of cardiovascular disease risk. Our findings align with those of other meta-analyses, which have also identified the cardiometabolic improvement effects of anthocyanin [32], catechin [33,34], curcumin [35], chlorogenic acid [36], flavanol [9], genistein [37], quercetin [38], and resveratrol [39].

Polyphenols offer several cardiovascular benefits, and these can be primarily attributed to a range of remarkable properties they possess. Among these are their antioxidant, anti-inflammatory, hypotensive, antihyperlipidemic, and hypoglycemic characteristics. The antioxidant property of polyphenols, which has been widely reported, plays a crucial role in cardiovascular health. This antioxidant ability is closely associated with the presence of hydroxyl groups within polyphenols’ molecular structure. These hydroxyl groups are of particular significance because they can be readily oxidized. When this oxidation occurs, it leads to the formation of the corresponding O-quinone. This chemical transformation is fundamental to the antioxidant mechanism of polyphenols and is directly related to their ability to counteract oxidative stress in the cardiovascular system, thereby contributing to the system’s overall health and proper functioning [14]. The transformation acts as an effective scavenger of reactive oxygen species. It operates based on the principle of entrapping free radicals within stable chemical complexes. By doing so, it halts the free radicals in their tracks and successfully averts any further reactions that these highly reactive species could initiate. This process is of great significance, as the unchecked reactions of reactive oxygen species can cause extensive damage to cells and tissues within the body, especially in the context of cardiovascular health, where such damage can have far-reaching consequences [40,41]. Beyond being able to suppress oxidative stress, polyphenols likewise practice indirect antioxidant effects through nuclear factor erythroid-associated transcription factor 2 (NFAT2) activation. The endogenous antioxidant system is stimulated by this process and may also be responsible for the maintenance of cellular redox homeostasis by polyphenols [41,42].

The anti-inflammatory properties of polyphenols are closely associated with oxidative stress and the regulation of cellular redox homeostasis [43]. The anti-inflammatory activity of polyphenols is mediated through several mechanisms, most of which involve suppression of the nuclear factor κB light chain enhancer in stimulated B cells. This minimizes the creation of pro-inflammatory mediators [44] and inhibits adhesion molecule expression [45], thus impacting monocyte chemotaxis in inflamed tissues. Polyphenols have also been demonstrated to enhance blood flow-mediated dilation within human subjects, despite their effective dose being considerably higher than that typically observed in dietary studies [46]. In vitro studies [47] and human studies [48] indicate that the mechanism by which polyphenols enhance blood flow-mediated dilation may be associated with an increase in nitric oxide synthase activity, which mediates vasodilatory effects and influences the renin–angiotensin system [49]. It is hypothesized that these effects contribute to the lowering of blood pressure, which is another mechanism by which polyphenols exert a cardioprotective effect.

Polyphenols possess a unique chemical structure and set of characteristics, which enables them to regulate cholesterol metabolism. This is achieved through the activation of the peroxisome proliferator-activated receptor-γ pathway, which in turn up-regulates the expression of HDL-associated farnesyl transferase paraoxonase 1 and increases its secretion. This process serves to prevent the peroxidation of LDL and HDL, while facilitating the efflux of cholesterol [50]. Furthermore, polyphenols diminish exogenous TG uptake by inhibiting pancreatic lipase activity, reduce fatty acid synthesis by decreasing fatty acid synthetase activity, and regulate cholesterol synthesis and efflux by modulating the sterol regulatory element binding protein gene and peroxisome proliferator-activated receptor α, thereby influencing blood lipid levels [51]. Polyphenols have been demonstrated to inhibit alpha-amylase and alpha-glucosidase, thereby inhibiting glucose uptake by sodium-dependent glucose transporter protein 1 in the gut. Additionally, they have been shown to stimulate insulin secretion and reduce hepatic glucose output [52]. Moreover, polyphenols facilitate insulin-dependent glucose uptake, stimulate the action of 5’-adenosine monophosphate-activated protein kinase, and alter the microbiome, collectively regulating glucose levels [53].

### Study Limitations

Multiple limitations should be noted while analyzing and interpreting the results of this meta-analysis. First, due to insufficient antioxidant-related randomized controlled trials, some antioxidant polyphenol supplements were not included in the present study. Second, evidence regarding improvements in the incidence of cardiovascular outcome events and T2D was absent in this study, making it inconclusive as to whether antioxidant polyphenol supplementation would also improve the incidence on these events. Third, several inconsistencies exist between our results and previous meta-analyses due to varying inclusion criteria. Fourth, notwithstanding the exclusion of trials involving interventional periods of inadequate duration (i.e., less than one week), heterogeneity among eligible studies was high due to limited numbers and sample sizes, variability in the intervention duration between studies, the broad time frame of the studies, and low-quality evidence for some of the heat-treated trials.

## 5. Conclusions

In conclusion, antioxidant polyphenol supplementation has been shown to benefit a range of cardiometabolic risk factors, including blood pressure, lipid profiles in the blood, and indicators of glucose metabolism, across the general population. The administration of antioxidant polyphenol supplements was observed to exert a range of beneficial effects on cardiovascular risk factors, irrespective of the participants’ underlying cardiometabolic health status. The findings we reported spotlight the significance of antioxidant polyphenol diversity and the necessity of balancing the benefits and risks for promoting and maintaining cardiovascular health. It is recommended that future research should focus on conducting more high-quality randomized controlled trials with longer follow-up periods. This will facilitate the evaluation of the long-term effects of antioxidant polyphenol supplementation on cardiovascular outcomes, including the incidence of CVD and T2D. Furthermore, studies should investigate the optimal doses and durations of supplementation for different polyphenols, with the aim of maximizing their beneficial effects while minimizing potential risks. Further investigations into underlying mechanisms of action are also warranted, with the objective of better understanding how these polyphenols exert their cardioprotective effects at the molecular and cellular levels. This knowledge could lead to the development of more targeted and effective intervention strategies for cardiovascular health.

## Figures and Tables

**Figure 1 nutrients-16-04206-f001:**
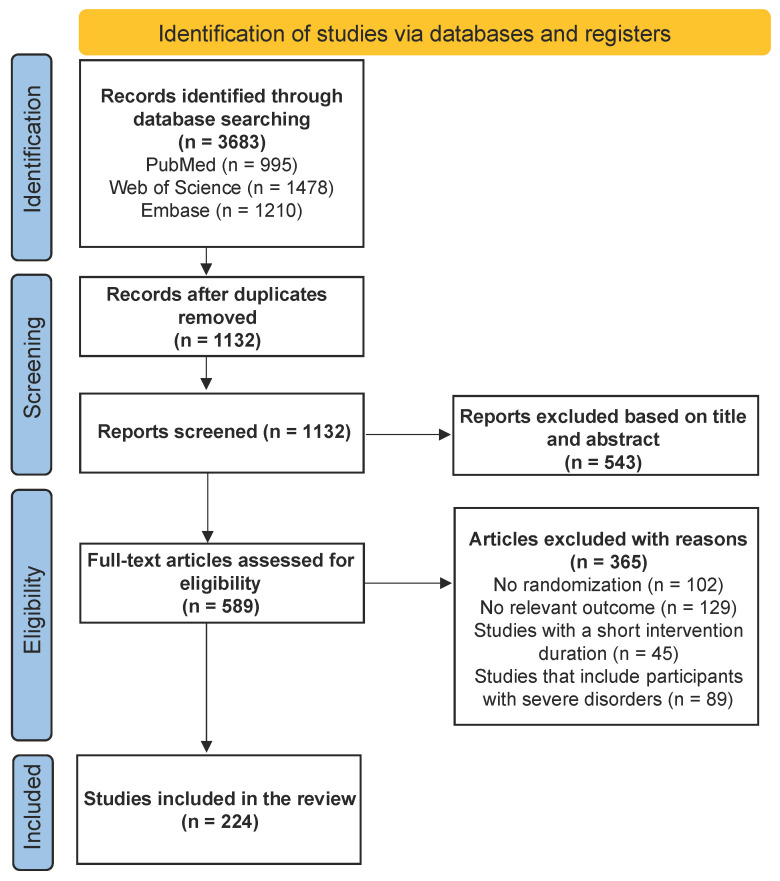
A flow chart illustrating the selection process for studies included in the study.

**Figure 2 nutrients-16-04206-f002:**
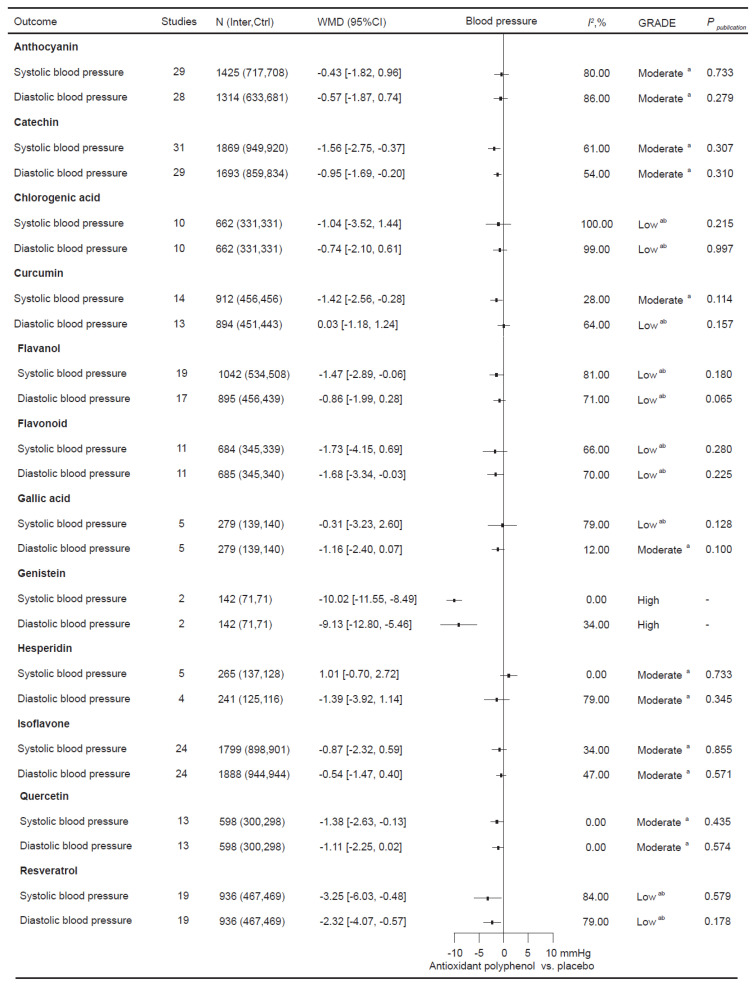
The objective of this study is to investigate the effects of antioxidant polyphenol supplementation on blood pressure. *I*^2^ considers whether the value of between-study heterogeneity is significant; ^a^ the rating was reduced due to the presence of imprecise elements; ^b^ the rating was reduced due to inconsistency.

**Figure 3 nutrients-16-04206-f003:**
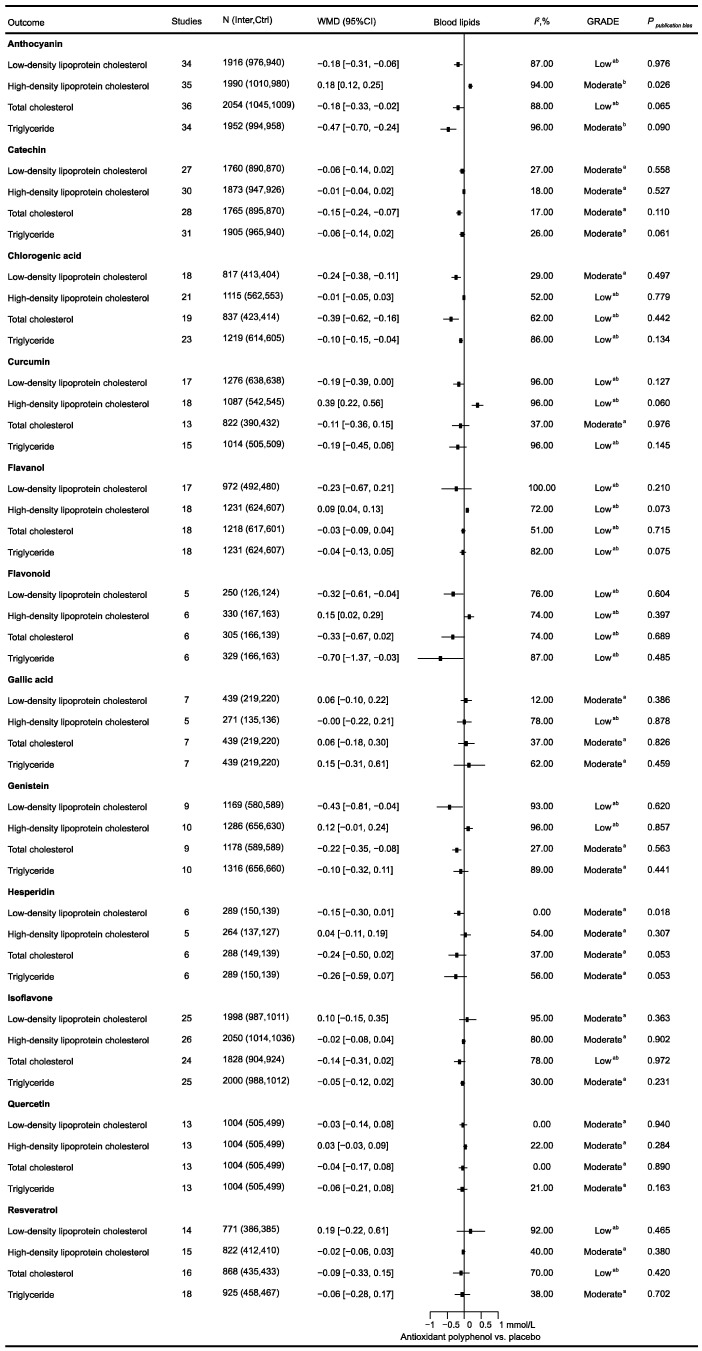
The objective of this study was to investigate the effects of antioxidant polyphenol supplementation on blood lipids. *I*^2^ considers whether the value of between-study heterogeneity is significant; ^a^ the rating has been reduced due to the presence of imprecise elements; ^b^ the rating has been reduced due to inconsistency.

**Figure 4 nutrients-16-04206-f004:**
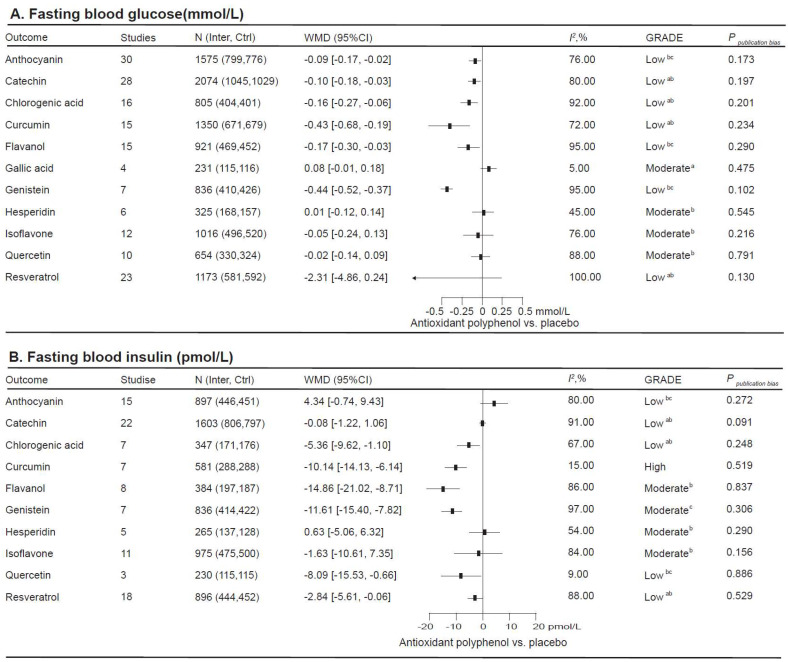
This study examines the impact of antioxidant polyphenol supplementation in relation to glycemic control. *I*^2^ considers whether the value of between-study heterogeneity is significant. ^a^ The rating has been reduced due to the presence of imprecise elements; ^b^ the rating has been reduced due to inconsistency; ^c^ the rating has been reduced due to risk of bias.

**Figure 5 nutrients-16-04206-f005:**
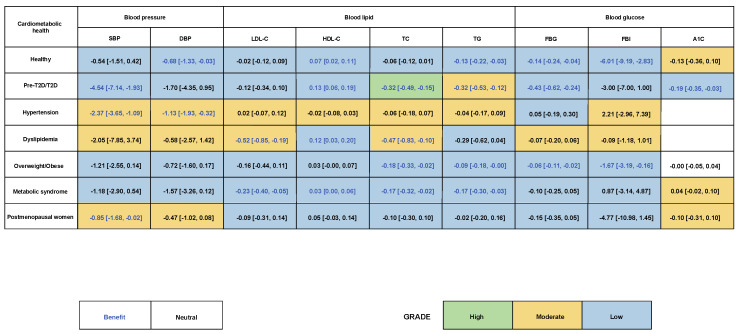
The aim of this study is to investigate the impacts of antioxidant polyphenol supplementation on participants with different cardiometabolic health conditions. A subgroup analysis was conducted to examine the effects of antioxidant polyphenol supplementation in two distinct groups: (1) individuals with optimal cardiometabolic health and (2) individuals with pre-type 2 diabetes (T2D) or T2D, along with those with hypertension, dyslipidemia, or metabolic syndrome, as well as those who are overweight or obese or postmenopausal women.

**Figure 6 nutrients-16-04206-f006:**
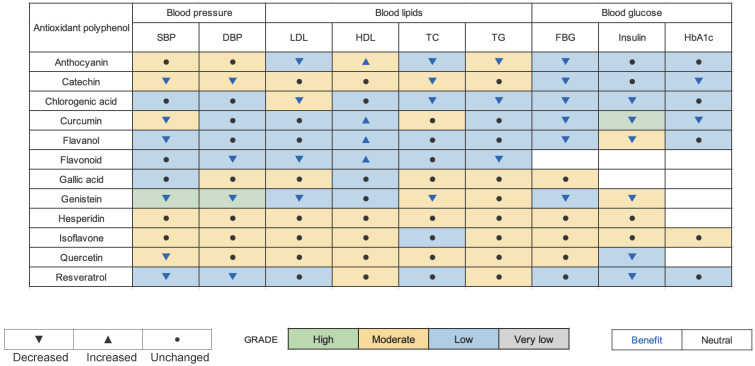
Antioxidant polyphenol supplementation for cardiometabolic health: An evidence-based map.

## Data Availability

The original data are available upon reasonable request. Requests to access the data must be to Yongting Luo.

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
