# Peer review of "The Effect of Antioxidant Polyphenol Supplementation on Cardiometabolic Risk Factors: A Systematic Review and Meta-Analysis"

_nutrients, 2024, doi:10.3390/nu16234206_

Round 1
Reviewer 1 Report (Previous Reviewer 3)
Comments and Suggestions for Authors
This systematic review aims to synthesize the existing evidence regarding the role of antioxidant polyphenol supplementation in the development of cardiovascular health. Authors concluded that antioxidant polyphenol supplementation has been shown to benefit a range of cardiometabolic risk factors, including blood pressure, blood lipids, and glycemic status across the general population. The reviewer considers the contents of this review to be useful clinical information from the perspective of preventive future cardiovascular disease. Furthermore, the reviewer considers that this systematic review thoroughly examines previous studies and summarizes it appropriately. The reviewer thinks that the current article has adequately addressed previous reviewer comments.
Author Response
Comments 1:This systematic review aims to synthesize the existing evidence regarding the role of antioxidant polyphenol supplementation in the development of cardiovascular health. Authors concluded that antioxidant polyphenol supplementation has been shown to benefit a range of cardiometabolic risk factors, including blood pressure, blood lipids, and glycemic status across the general population. The reviewer considers the contents of this review to be useful clinical information from the perspective of preventive future cardiovascular disease. Furthermore, the reviewer considers that this systematic review thoroughly examines previous studies and summarizes it appropriately. The reviewer thinks that the current article has adequately addressed previous reviewer comments.
RESPONSE 1: Thank you for your positive feedback on our manuscript. We appreciate your recognition of the value of our study and the thoroughness of our review. We will keep up the highest scientific integrity standards and ensure our research findings are reliable and practical.
Reviewer 2 Report (Previous Reviewer 1)
Comments and Suggestions for Authors
I suggest the authors to improve their manuscript according to the following comments:
References are not formatted according to the journal’s guidelines.
Line numbering is missing.
“Overall, our results suggest in the overall population, genistein, anthocyanin, catechin, chlorogenic acid, curcumin, flavanol, flavonoid, resveratrol, and quercetin notably enhanced various cardiovascular risk markers, including hypertension, lipid profiles in the blood, and indicators of glucose metabolism” – Which results? You don’t mention any results in the abstract. In this section, conclusions and directions for further actions and investigations are missing.
The Introduction needs to be improved and expanded. What is the justification to perform this review and what does it bring new to the current knowledge on the topic? What is the novelty factor?
Provide more details in section 2.3.
In the Results section, you mention that 224 were included in your review. Isn’t this too much? How do you explain it and how do you analyze all these studies?
Improve the quality of Figure 5.
Align your Conclusions with the improved and reformulated abstract.
Author Response
Comments 2: References are not formatted according to the journal’s guidelines.
RESPONSE 2: We apologize for this mistake. The proper formatting of references has been corrected in the manuscript.
Comments 3: Line numbering is missing.
RESPONSE 3: Line numbering has been added in the manuscript.
Comments 4: “Overall, our results suggest in the overall population, genistein, anthocyanin, catechin, chlorogenic acid, curcumin, flavanol, flavonoid, resveratrol, and quercetin notably enhanced various cardiovascular risk markers, including hypertension, lipid profiles in the blood, and indicators of glucose metabolism” – Which results? You don’t mention any results in the abstract. In this section, conclusions and directions for further actions and investigations are missing.
RESPONSE 4:
Thanks for your suggestion. We understand your concern regarding the lack of specific results in the abstract. However, we would like to present our perspective on this matter.
The abstract is intended to serve as a concise summary that provides a quick overview of the study's key objectives, methods, and overall conclusions. Given the complexity and the large number of results in our study, including a detailed account of all the findings in the abstract would make it overly long and potentially detract from its main purpose, which is to give readers a clear understanding of the study's significance and main contributions at a glance. We have presented the extensive results in a comprehensive and detailed manner in the main body of the paper. The main focus of our research is on the synthesis and meta-analysis of the collective evidence rather than on presenting individual results in isolation. We believe that the detailed results are better suited for in-depth exploration within the context of the full manuscript, where readers can access the complete analysis and discussion.
Nevertheless, we understand the importance of providing some indication of the results in the abstract. To address this, we are willing to consider including a brief summary or a few key examples of the most significant results. However, we would like to caution that including too many detailed results in the abstract might compromise its clarity and make it more difficult for readers to grasp the main message of the study. We hope that you can understand our position and that this alternative approach will satisfy your concerns while maintaining the integrity and readability of the abstract.
“Our results suggest multiple polyphenol supplementation improved cardiovascular risk markers in the overall population. For example, catechin supplementation decreased systolic (-1.56 [-2.75, -0.37] mmHg)and diastolic blood pressure (-0.95 [-1.69, -0.20] mmHg). Anthocyanin supplementation improved multiple blood lipid profiles. Curcumin supplementation benefits indicators of glucose metabolism.”
Besides, in this section, conclusions and directions for further actions and investigations have been added.
“Our meta-analysis provides comprehensive evidence that antioxidant polyphenol supplementation can have beneficial effects on various cardiometabolic risk factors in the general population. The observed improvements in blood pressure, lipid profiles, and glycemic status support the potential role of these supplements in cardiovascular health promotion. However, the heterogeneity among studies indicates that more research is needed to fully understand the optimal use of different polyphenols. Future research should concentrate on conducting a greater number of well-designed randomized controlled trials over extended periods of time to evaluate the long-term impact on cardiovascular endpoints and to ascertain the optimal doses and durations of antioxidant polyphenol supplementation. Furthermore, additional research is required to gain a deeper understanding of the underlying mechanisms responsible for these cardioprotective effects.”
Comments 5: The Introduction needs to be improved and expanded. What is the justification to perform this review and what does it bring new to the current knowledge on the topic? What is the novelty factor?
RESPONSE 5: The introduction has been improved and expanded.
“Despite the existing evidence suggesting the potential benefits of polyphenols on cardiovascular health, there are still significant inconsistencies in the literature. It is of great clinical importance to gain a deeper understanding of the impact of antioxidant polyphenol supplementation on cardiometabolic risk factors. If proven effective, polyphenol supplements could represent a relatively inexpensive and accessible preventive or adjunctive treatment option for cardiovascular diseases. The objective of our review was to synthesize the available evidence in a more comprehensive and systematic manner, with the aim of addressing the aforementioned uncertainties and providing a more definitive understanding of the role of antioxidant polyphenol supplementation in cardiovascular health.
To pool the available evidence regarding the role of antioxidant polyphenol supplementation in advancing cardiovascular health, a systematic review and meta-analysis was conducted that included all available randomized controlled trials of antioxidant polyphenol supplementation and its effect on CVD risk factors. This comprehensive approach provides a more holistic view of the potential benefits of antioxidant polyphenols as a class of compounds. The impetus behind conducting this review was to provide evidence-based recommendations that could potentially be translated into clinical practice and public health strategies with the aim of improving cardiovascular health outcomes.”
Comments 6: Provide more details in section 2.3.
RESPONSE 6: More details have been provided in section 2.3.
Comments 7: In the Results section, you mention that 224 were included in your review. Isn’t this too much? How do you explain it and how do you analyze all these studies?
RESPONSE 7:
We acknowledge your concern regarding the relatively large number (224) of studies included in our review. We recognize that this may initially appear overwhelming. However, we would like to provide an explanation of the rationale behind this approach and the methods employed to handle such a substantial amount of data.
There is a wide variety of antioxidant polyphenols, each with potentially different effects on cardiometabolic risk factors. To comprehensively assess the overall impact of this class of compounds, it was necessary to include a significant number of studies that investigated different polyphenols, such as genistein, anthocyanin, catechin, and many others. This breadth of coverage allows for a more comprehensive understanding of the potential benefits and variations within the group of antioxidant polyphenols. Given the complexity of the topic and the different experimental designs, populations, and intervention protocols used in previous research, there is substantial heterogeneity among studies. By including a larger number of studies, we were able to capture and analyze this variability more effectively. This approach provides a more representative picture of the real-world evidence and helps to identify patterns and trends that might not be evident with a smaller sample of studies. A larger sample of studies generally leads to more robust and reliable results. It reduces the impact of individual study biases and increases the statistical power of our meta-analysis. This is particularly important in a field like ours, where the effects of antioxidant polyphenols on cardiometabolic health are still being explored, and small or inconsistent effects might be overlooked with a limited number of studies.
We followed the Preferred Reporting Items for Systematic Reviews and Meta-Analyses (PRISMA) guidelines meticulously throughout the review process. This involved a detailed and structured approach to study selection, data extraction, and analysis. Two independent reviewers screened all the identified studies at each stage (title/abstract and full text) to minimize selection bias. Any disagreements between the reviewers were resolved through discussion and, if necessary, by consulting a third reviewer. This rigorous process ensured that only eligible and relevant studies were included in the final analysis. For each included study, we extracted a wide range of data using a pre-defined data extraction form. This form included details such as study characteristics (author, publication date, study design), participant characteristics (age, gender, health status), intervention details (type and dose of polyphenol, duration), and outcome data (changes in blood pressure, blood lipids, and glycemic status). By standardizing the data extraction process, we ensured consistency and accuracy across all studies.
To analyze the data from the included studies, we used appropriate statistical methods. For continuous outcomes (such as blood pressure, lipid levels, and glucose metabolism indicators), we calculated weighted mean differences and 95% confidence intervals using a random effects model. This model takes into account the heterogeneity between studies, which is expected given the diversity of the included studies. We also assessed the heterogeneity using the I2 statistic, which provides an estimate of the proportion of the total variation in the effect estimates that is due to between-study differences. In addition, we conducted subgroup analyses to explore whether the effects of antioxidant polyphenols differed depending on various factors such as the type of polyphenol, the health status of the participants, and the duration of the intervention. To evaluate the potential for publication bias, we used Egger's regression test and visually inspected funnel plots. Sensitivity analyses were performed by sequentially excluding each study to assess the stability of our results and the influence of individual studies on the overall findings.
Comments 8: Improve the quality of Figure 5.
RESPONSE 8: The quality of Figure 5 has been improved.
Comments 9: Align your Conclusions with the improved and reformulated abstract.
RESPONSE 9: Conclusions has been aligned with the improved and reformulated abstract.
“In conclusion, antioxidant polyphenol supplementation has been shown to benefit a range of cardiometabolic risk factors, including blood pressure, lipid profiles in the blood, and indicators of glucose metabolism across the general population. The administration of antioxidant polyphenol supplements was observed to exert a range of beneficial effects on cardiovascular risk factors, irrespective of the participants' underlying cardiometabolic health status. The findings we reported spotlight the significance of antioxidant polyphenol diversity and the necessity to balance benefits and risks for promoting and maintaining cardiovascular health. It is recommended that future research should focus on conducting more high-quality randomised controlled trials with longer follow-up periods. This will facilitate the evaluation of the long-term effects of antioxidant polyphenol supplementation on cardiovascular outcomes, including the incidence of CVD and T2D. Furthermore, studies should investigate the optimal doses and durations of supplementation for different polyphenols, with the aim of maximising their beneficial effects while minimising potential risks. Further investigations into the underlying mechanisms of action are also warranted, with the objective of better understanding how these polyphenols exert their cardioprotective effects at the molecular and cellular levels. This knowledge could lead to the development of more targeted and effective intervention strategies for cardiovascular health.”
This manuscript is a resubmission of an earlier submission. The following is a list of the peer review reports and author responses from that submission.
Round 1
Reviewer 1 Report
Comments and Suggestions for Authors
The similarities with other work already published by the same authors are very evident: https://www.mdpi.com/2072-6643/16/14/2213
A total of 53% similarity index is not admissible. Furthermore, it is necessary to take into account that 30% correspond to similarities with the work mentioned previously.
The novelty factor of this work is non-existent, therefore it should not be considered for publication in the journal Nutrients.
In my opinion, submissions like this should be directly rejected, as reviewers can't evaluate a work with these characteristics and where plagiarism is too evident.
Reviewer 2 Report
Comments and Suggestions for Authors
There found to be several points to be reconsidered through the Editorial check.
Reviewer 3 Report
Comments and Suggestions for Authors
This systematic review aims to synthesize the existing evidence regarding the role of antioxidant polyphenol supplementation in the development of cardiovascular health. Authors concluded that antioxidant polyphenol supplementation has been shown to benefit a range of cardiometabolic risk factors, including blood pressure, blood lipids, and glycemic status across the general population. The reviewer considers the contents of this review to be useful clinical information from the perspective of preventive future cardiovascular disease. Furthermore, the reviewer considers that this systematic review thoroughly examines previous studies and summarizes it appropriately. However, there are several questions in this study.
As the authors explain in the “Introduction” section, polyphenols are typically classified into flavonoid and nonflavonoid types. Based on the content of this study, the reviewer infers that the authors conducted a search for randomized controlled trials (RCT) and performed meta-analysis focusing on flavonoid type. In the “Introduction” and “Methods” sections, the authors should explain the significance of focusing on flavonoid type.
In this meta-analysis, the authors extracted data from the cited RCTs on study design, participant characteristics (number of participants, mean age, sex, and health status), intervention substance, control substance, intervention dose, and intervention duration. The reviewer believes that these factors are extremely important information when interpreting the results of this study. The reviewer considers that if this information could be summarized in supplement files, this study would be of higher quality.
In this study, the authors examined the effects of each polyphenol on cardiometabolic risk factors. However, in this study, the authors only examined the effect volume of supplementation with each polyphenol. The reviewer also thinks it important to consider whether there is a dose-response relationship between supplementation with each polyphenol and improved cardiometabolic risk. The reviewer believes that the study would be of higher quality if trend tests were added to each polyphenol to determine whether there is a dose-response relationship between the effects of supplementation with each polyphenol and the improvement in cardiometabolic risk.
The sample used in this study included participants aged between 18 to 75 years (median age: 52.2 years). Did aging have an effect on the improvement of cardiometabolic risk with antioxidant polyphenol supplementation? The reviewer thinks that the study would be of higher quality if the effects of aging could be added to the results in Figure 5.